# The Impact of COVID-19 on Purchase Behavior Changes in Smart Regions

**Mária Pomffyová *** and **Lenka Veselovská**

Institute of Managerial Systems in Poprad, Matej Bel University, 05801 Poprad, Slovakia
* Correspondence: mpomffyova@umb.sk; Tel.: +421-908069333

**Abstract:** The COVID-19 pandemic has changed consumer behavior due to various restrictions and increased degrees of ICT use. By establishing and verifying the validity of the hypotheses, we aim to compare intensities of mutual correlations that indicate changes in consumer behavior depending on the degree and nature of changes in selected socio-demographic or socio-economic factors. The statistical evaluation of the answers obtained in surveys of representative samples of 987 respondents from the Slovak Republic (implemented in 2021 about the dual quality of goods sold in the EU) and also the answers of 347 respondents (in 2022 aimed at changes in Slovak consumer behavior) will be carried out with multivariate analyses using the SPSS program. The outputs indicated that during self-isolation periods, Slovak consumers bought more or the same amount as before the pandemic; shopping habits were mainly changed by women and groups with lower household income. Test subjects preferred the quality products and products posing the least amount of risk to health. All consumers intend to continue to shop through e-commerce platforms where they prefer a more personal experience (through social media or YouTube). Low-income people's budgets are threatened by cheap products and poor distribution of spending, especially among young people. We recommend simplifying personalized visualized sales and education content and e-methods of information sharing also in order to make them accessible to digitally disadvantaged groups (according to income, age, education, etc.). The use of blockchains increases transparency of production and sales value chains, reducing the occurrence of unfair practices, and promoting participatory public dialogue.

**Keywords:** ICT; smart technologies; consumer behavior; generational factors; e-identity





## 1. Introduction

As the impact of the COVID-19 crisis has shown, both organizations and individual consumers need ICT (Information and communication technologies) to conduct their everyday activities. The spread of social interaction and shopping in a virtual environment as well as the realization of payments in combination with goods delivery to home becomes a reality and a matter of course. The increasing number of possibilities of e-connectivity forms a prerequisite for a proactive approach to the digital economy. As is known, their use made people's lives easier even before the pandemic [1–3].

Based on research of consumer behavior and its possible changes, new trends can be observed in various world regions [4]. The need for isolation and an increased level of self-security has made it a trend to shop for food, clothing or other necessities more frequently via e-commerce platforms. The disadvantage is that the consumer cannot provide or receive the information about the goods as can be done in person. The greatest attention is paid to the area of food safety and catering [5], where the issue of dual food quality has already become important in the past. Food safety is considered a part of public wellbeing, and any form of mismanagement by its producer can cause serious harm to all those involved in the supply chain, adversely affecting public health. This issue is also invoked by manufacturers who argue that they comply with certain rules and regulations

in the production processes, be it at national or international level [6]. Despite the crisis, manufacturers are increasingly using unfair practices. Transparency of production, delivery and sales processes are becoming more and more important, which is not always the case; therefore, attention needs to be paid to this issue.

Consumers have to be well-informed, which increases the need for technologies offering the ability to obtain information online, where, for example, blockchain technologies originate [7]. During the lockdown, the importance of social media and networks, as well as the role of the family, community or close surroundings increased [8]. Regardless of age, especially people in the workforce had to learn to adapt to specialized equipment or services that require employment of advanced digital skills.

The generation of millennials (adults aged 20–40) is the generation that has a favourable attitude towards communications, media and digital technologies [9] and masters the tools that previous generations find challenging to implement. They are familiar with social media, which also influences their consumer behavior, and they can adapt quickly to new processes. In this sense, the marginalized groups are mainly people of productive age or older people who adapt to working with specialized tools or complex services much more poorly [10]. Therefore, it is important to know their preferences and help make the acquisition of digital skills as easy as possible.

The use of online connectivity and available services is compromised by community size and broadband coverage, as well as different user skills depending on their gender or income. According to comparison of the indicators of Eurostat Statistics from 2018 to 2022, we can conclude that due to the impact of the COVID-19 pandemic, the digital gap (approx. $-2\%$) of respondents in the distribution by region or size of residence has decreased. The positive impact on the digital literacy of adults in Slovakia was also confirmed by our previous research results, as well as the research results of the Institute for Public Issues (IPA) focused on public issues and transformation processes of companies [11]. More than two thirds of respondents (survey conducted in February 2022 on a sample of 1003 respondents older than 18 years in SR) confirmed enhancement of their basic skills, but only 41% confirmed progress in the use of specialized, possibly professional applications. People with enhanced skills or those who prefer ICT use, regardless of their place of residence or level of education, reported a lower degree of their skills' improvement. It is necessary to emphasize this, even though according to [12], the generation of millennials (adults aged 20–40) will account for the largest share of buyers in the market by 2030.

The question remains: To what extent were these changes caused by the pandemic itself as opposed to the consumers' reluctance to learn something new and subsequently develop their own digital skills? Another relevant question is the role of the companies' ability to quickly adapt to the conditions of the pandemic, and the extent to which the government measures were proactive, whether with consumers, manufacturers or suppliers. Since the development in this area is taking place very quickly, there is a lack of surveys in the scientific or professional literature. The aim of this paper is to examine the ways in which various socio-demographic or socio-economic factors influenced changes in consumer behavior and whether it is possible to increase the transparency of value chains. In the first part, we analyse theoretical issues, summarize gaps and set research hypotheses. Next, we present datasets and research methods. In the next part, we interpret a summary of empirical results that either confirm or deny our hypotheses. In the conclusion, we summarize the findings, research limitations and future research activities.

## 2. Literature Review

### 2.1. Impact of Self-Isolation on Consumer Behavior Changes

The COVID-19 pandemic has changed market conditions over the course of its duration, and this trend continued in 2022. Even though during spring 2020, Slovakia was among the countries least affected by the pandemic, in 2021, the situation drastically worsened. However, compared to other countries, the measures implemented by the Slovak government were considered to be strict.

Next, other negative external factors have started to influence the consumers in Europe, mainly the increasing inflation, energy crisis or armed conflict, due to which the pandemic ceased to be considered as major focus of our lives. Nevertheless, in the Slovak Republic, as well as in other countries, the rate of consumption increased more slowly (for example, in China, among others, where the country's top leadership considers boosting consumer spending an economic priority in the five-year plan leading up to 2025, as well as the nation's long-term vision through 2035 [13]). The reason for this is the fact that rural inhabitants were interested in compromising their spending and implementing other forms of savings to accumulate emergency savings. As stated in [14], the pandemic caused a decrease in savings in India and also reduced Singapore household spending on consumption by almost a quarter compared to the peak in 2020 [15]. It may be related to the rising unemployment rate, declining labour productivity, and worsening income stability [16]. Likewise, the authors of [17] discovered that income uncertainty in Italy contributed to savings increase and consumption drop.

Even though there are no similar research studies focused on Slovak consumers enabling a comparison of findings, the situation in the Slovak Republic is almost the same. Especially alarming is the discovery that more than a quarter of Slovak consumers spend more than 90% of their household income on buying goods and services, leaving less than 10% for savings [18]. It is assumed that due to other adverse events in the regions, for example, close proximity to a country in conflict or growing inflation, will even worsen the bad financial situation.

The next most common reason for changing purchasing habits and the types of products was their availability. Companies in Slovakia adapted quickly (confirmed in research by KPMG [19]), and new e-shops were created during 2020 as a way to expand sales channels. This subsequently caused changes in shopping habits of various specific groups of Slovak consumers.

Therefore, we want to examine the change in the volume of Slovak purchases due to the mandatory self-isolation, and the perception of these changes in shopping habits by the population, classified into groups by gender and income. We proposed the following hypothesis:

**Hypothesis 1:** *The amount of income and unavailability of products and stores during self-isolation due to pandemic are decisive factors that significantly change consumer behavior and the amount of purchased quantity of lowest income groups.*

### 2.2. Healthy Lifestyle and Quality of Products

Impact of pandemic restrictions on mobility and fears of infection caused the growing emphasis on a healthy lifestyle, which is also related to the active spending of free time. Decreased quality of logistics pertaining to goods or their unavailability caused the wider use of local resources (material or labour), which also supports sustainable development of the regions [20]. Such changes can be evaluated as a positive impact of COVID-19, where the following resulting consumer preferences were identified:

- to spend free time in the spirit of "returning to nature",
- to require quality at all levels (quality of water, services, food, etc.).

According to our research findings (research activities in 2021), it is difficult to clearly identify the dependence of inclination to lead a healthy lifestyle on sex [21] and subsequent change in consumer behavior. While men considered price a deciding factor, women tended to focus on product quality. It is understandable, however, that at a time when health is at risk, women begin to prioritize quality and healthy products, not only for themselves but also for other household members with an effort keep the whole family in good shape.

It is becoming a trend to prepare food at home, either because of the high price of pre-made meals, concern about the quality of the food, or the low financial contribution of the employer to food budget [22]. This fact is also confirmed in research outputs on 2000 consumers in Germany, described in [23], where more than 62.9% of consumers claim that they prepared meals at home before the crisis, and more than one-fourth of them

indicated they would continue to do so due to fear of infection risk in public place. In addition, in Slovakia, approximately 54.9% of respondents variable stated that they have to budget for food, even of lower quality. As service providers must adapt to this trend in order to maintain the satisfaction of visitors of destinations and keep them coming back, they have to be aware of the respondents' preferences.

The need to increase interest in product quality as opposed to price requires the knowledge of consumer preferences in the area of price and cost control. However, in addition to legislation and ethics, quality also has an economic dimension; in academic research, the necessary attention is lacking on the topic, as price comparison is usually not the primary goal of dual quality of goods testing [24]. According to several tests, goods purchased in Western European countries not only had a healthier composition, but also were cheaper in the same package of the same weight (or priced the same but with a higher weight) [25].

There are certain restrictions related to the question of income or prices of goods. As the existing income gap in Central and Eastern Europe vis-à-vis Western Europe creates a sense of unfair position of the countries in the EU single market, it is essential to pay attention to the issue of prices and costs. It cannot be claimed that an expensive product will also be of high quality or less hazardous to health. There exists some limitation. We will examine the preferences of the respondents, but they cannot always be generalized.

In order to find out the effect of income on the change in consumer behavior due to the change in the quality or price of products, we established a hypothesis:

**Hypothesis 2:** *The lower the income, the lower the interest in quality and health risk.*

The interest in ways to protect oneself from health risks has always existed. In the past, several tests have been performed to confirm the existence of dual quality in several countries [24], but a uniform testing methodology was lacking. In 2017, at the initiative of the Minister of Agriculture of the Slovak Republic at the Council of the EU, the fight for consumer rights began, which resulted in the inclusion of the dual quality of foods in the list of Unfair Commercial Practices. The European Commission's Joint Research Centre (JRC) developed an identical methodology for comparing product quality and composition and launched its dual quality testing in 2018 to analyse 1380 food products in 19 EU countries (also mentioned by [26]). Subsequently, in 2019, the testing of products other than food was launched. The directive on identical methodology was issued in 04/2019 by the European Commission and individual member states began to apply it to their legislation. The European Parliament responded to this situation approving the Directive (EU) 2019/2161 on better enforcement and modernization of Union consumer protection rules as a part of "New Deal for Consumers". The aim was to ensure legal clarity in the assessment of possible cases of dual quality, to strengthen the powers of consumer protection authorities and also the possibility of imposing sanctions on producers (up to 4% of turnover) [27].

According to the European Commission, products may have different consistencies in different countries if manufacturers state this in the formulation on the product. Thus, it is not a legislative problem, but an ethical problem, because if consumers buy the same product in different places, they assume that it is the same [28]. The issue of dual food quality has three planes [29]. The first level is product safety. Although the composition of products in some countries is different in its components, it does not matter, because the products are still safe and are made from raw materials that are legally approved for consumption in that country. If different raw materials are allowed in different countries, manufacturers will adapt to this regulation and choose the most advantageous option for them (often cheap, but less tasty, etc.). At the same time, manufacturers state that they provide all the information on the packaging, thus also complying with all regulations.

The financial crisis increases the risk of unfair practices. It is more than necessary to provide an overview of them, since consumer protection is essential. In the context of food

safety in the EU, a Rapid Alert System (RAS) is implemented, focusing on, for example, the presence of foreign objects in food or the concentration of food supplements. The feeling of Central and Eastern European consumers that they are a "rubbish bin" is therefore partly justified (as a 2017 survey on a sample of 1446 samples showed, most of the contaminated food registered in the RAS system originated from Eastern Europe [24]). In food safety measures, the basic management tool for monitoring compliance and gaining customer confidence is the introduction of a food safety management system (FSMS) according to the international standard ISO 22000: 2018, valid after 29 June 2021 [30]. It also integrates the principles of the Hazard Analysis Critical Control Points (HACCP) system and its application steps. Adherence to this system supports more effective control of processes, employees, suppliers, raw materials, and final products. The external benefits are gaining consumer confidence in food safety and improving the company's image in the eyes of the consumer and control authorities.

In Slovakia, the State Veterinary and Food Administration of the Slovak Republic and its regional authorities (RVPS) are responsible for food quality and safety control. They carry out supervision and inspections, regularly issuing reports on the matter (monthly and occasionally). They also have the power to receive suggestions, complaints or petitions from various bodies, as well as to provide information. Their admission is organized in the form of a web interface with the possibility of any form of submission: written, oral (documented in writing), or through the electronic filing office—the www.slovensko.sk (accessed on 17 May 2022) portal, and in the case of suggestions also in the electronic form [31]. The problem is whether customers want to participate electronically. It is assumed that the level of knowledge about product quality issues and unfair practices will increase the willingness to participate. At the same time, the question is whether reporting applications are user-friendly. To explore this, we proposed the following hypothesis:

**Hypothesis 3:** *The perception of the need for participation in solving the problem using media depends on the level of knowledge of unfair practices.*

*2.3. The Role of Media as a Tool to Be Well Informed and the Need for Soft Skills*

Regarding the obtaining of the information, the vulnerable group, which was especially evident during the COVID-19 crisis, were seniors who suffered the most from isolation. Digital connectivity was inaccessible to many of them, either because they did not have technical devices or did not know how to use them. Therefore, it is necessary to examine their trusted sources of information and their preferred methods of obtaining the information.

The impact of social media is enormous due to the technological development in health care. Some of the faculty (teaching) hospitals and private institutions are already using online ordering forms to communicate with patients or social media to increase awareness and to provide medical education to the public. In addition, Slovak government intends to support online electronic healthcare as a part of its Recovery plan [32]. Online Health Services and their acceptance, especially among the older generation, requires the need to build an appropriate relationship with them. Since searching for essential medical information, booking doctor's appointments, obtaining the results of medical examinations online and even remote telemedicine services are more than challenging for this generation and often available only to a small group of senior citizens [33], the family or the community play an important role of an advisory body while providing support for all online health-related activities.

In this regard, it is necessary to examine which environment is most suitable as a communication channel in marginal groups. We proposed the following hypothesis:

**Hypothesis 4:** *Changes in shopping habits depend on the intensity of one's own experiences and community more than on those reported by the media.*

### 2.4. Digital Economy and Its Impact on Consumer Behavior

For businesses, it is always difficult to quickly adapt to the aftermath of crisis events. Due to various external influences (such as technological or contextual changes, local market opportunities or third-party partnerships in the consumer value development), business models changed, requiring innovation factors to make their natural basis. However, the lack of research increases the need to search for new solutions that will contribute to simplification of various communication, shopping, business or other activities while simultaneously reducing the energy demand and ensuring savings for the groups involved.

The aforementioned reasons for conducting business in virtual environment was mentioned by almost a fifth of respondents [34], when during the pandemic up to a quarter of Slovak e-shops were created. The research results from abroad also showed that the rate of relocation to the online trading during the pandemic has been significant [35–37]. Businesses often adapted to this situation by creating a website, or adjusted the current one so that customers could order their goods through it. Because people are shopping in stores less, sellers have to offer both a physical and digital presentation of a store to meet all customer demands on any platform. Consumers can then pick up their goods and pay at the dispensing point in the area of the stone store in compliance with all applicable current anti-pandemic measures, or their goods are delivered to their door overnight.

E-commerce created new economic and social relationships among both enterprises and individual consumers based on the broadened use of ICT that became a common need of digital transformation [38]. Traditional civil agreements can be replaced by smart contracts, also called self-executing or digital, which represent a new type of contract relations using blockchain technology. Their advantage is guaranteeing confidentiality and confirming e-identity of civil relations. The disadvantage is that there are also some shortcomings resulting from an automated algorithm which can cause financial loss, reducing trust in such tools. These deficiencies can be eliminated by auditing codes by reputable IT professionals. Blockchains cannot completely replace human behavior, but they support the independence of finance from the traditional financial and political system [7].

As product prices increase due to inflation, manufacturers often try to reduce their costs, but they also use unfair practices. In turn, as consumers are increasingly demanding transparency and traceability in the production of food and beverages and the documentation of these production processes, blockchains offer the opportunity for transparency of these processes. Their use requires expanding the functioning system, but also users' skills and knowledge enhancement for those with advanced or expert skills that are still lacking. In any case, the transition to online trading in some form is essential for businesses, as this change in consumer behavior is proving to be long-lasting and is likely to persist in the future [39].

The question is whether consumers are willing to change their shopping habits even in the post-pandemic period, given the related restrictions when purchasing or demands for digital skills. We proposed the following hypothesis:

**Hypothesis 5:** *Changes in shopping habits depend on consumer gender and income group.*

### 2.5. The Role of Smart Solutions and Innovation
#### 2.5.1. Artificial Intelligence and Innovation

In recent years, changes in demography and social structures have caused a higher rate of participation of women in the labour market. In Western civilization, the role of the woman in the household is changing, resulting in the associated professional emancipation and increased share in the decision-making role, which also affects changes in the consumers' preferences. Businesses are forced to adapt and quickly respond to various situations; making business decisions based mostly on intuitive evidence, past performance evaluation, or instincts is not enough. Those business leaders who use a data intelligent technology will be more competitive and achieve better business results. Data drives better strategic business decisions.

In addition, changes in the way of selling require to reduce manual processes and redundant work. Employees must be able to perform more accurate analyses; in this regard, artificial intelligence (AI) has its place. Many business processes can be automated, but when digital transformation is complicated and increases costs, it is necessary to use the experience, knowledge, and tools of specialists [19]. Cooperation with experts helps to incorporate intelligent automation into business processes which become more efficient, and at the same time employees expand their experience and their work has a higher added value. Potential collaboration supports better IT transformation that improves automation manufacturing and regulatory measures across worldwide locations, which leads to savings in IT finance and enhancement of positive approach to research and development [40]. In addition, all these changes require the development of advanced specialised skills in all those who use the services or tools supported by ICT.

### 2.5.2. Smart Solutions as a Way of Innovation Support

Intelligent solutions require the development of various research or testing capacities and the exchange of best practices between the actors involved. In [41], the authors propose that the application of smart technologies can increase the sustainability of business activities, provided that there is an active support by coherent regional policies. Start-up solutions and SMEs (small and medium enterprises) have innovative potential resulting in intelligent solutions, but their finances are limited. The formation of consortiums between financial institutions and other legal, technological and consulting companies is necessary [42,43]. As wider public dialogue supports exports and investments, economic diplomacy, digitization and education, the Ministry of Economy of the Slovak Republic formed the Slovak Government Council for Competitiveness and Productivity [44] with the aim to support SMEs regional innovation potential to produce smart solutions that arise as a common need.

In Slovakia, FinTech Hub Slovakia was the first open platform until 2020 with the aim to support start-ups of electronic solutions for financial management [45]. Support for solutions such as payment with smart portable devices, facial biometrics, or smart payment cards helps to simplify and reduce the costs for customers and also companies (financial, personal or operational).

The Smart Cities Action Plan is aimed at the enhancement of city intelligent ecosystems where development of the market for intelligent and innovative solutions has its place. In 2022, in the Plan for Intelligent Industry of the Slovak Republic (granted by the Ministry of Economy of the Slovak Republic), the conceptual support was divided into four areas: data, IoT, Innovations, Participation and business environment, also presented in [46]. The gap in this area is the fact that financial support of innovations in City concepts is not governed by a unified concept; their development is non-conceptual.

We researched the general awareness of these concepts and their possibilities to develop support of e-participation with the public sector. We found out that out of 40% of large cities of the Slovak Republic and 100 representatives of companies that cooperated in the Modernization of Local Territorial Self-Government within the Operational Program Effective Public Administration, approximately 1/5 cities did not implement smart solutions. In 2022 (compared to 2020), there was an increase from 15% to 40% in the number of large cities collaborating on smart solutions.

Figure 1 shows the areas of smart solutions and funding schemes in the form of different operational programmes [47]: Smart Energy (SE), Quality environment (QE), Intelligent City Infrastructure (IROP, OP QE, and OP II), Smart Economy (R&I), Intelligent Services (OP EVS, OP II), Intelligent Transport, Intelligent Housing, Education and Social insurance (OP HR, OP II).

City representatives are willing to invest in innovations (96%), mostly in transport (its management, parking and e-mobility), smart city services (social services, communication, security, etc.). However, the lack of investment and experts in the city management is a basic obstacle [48].

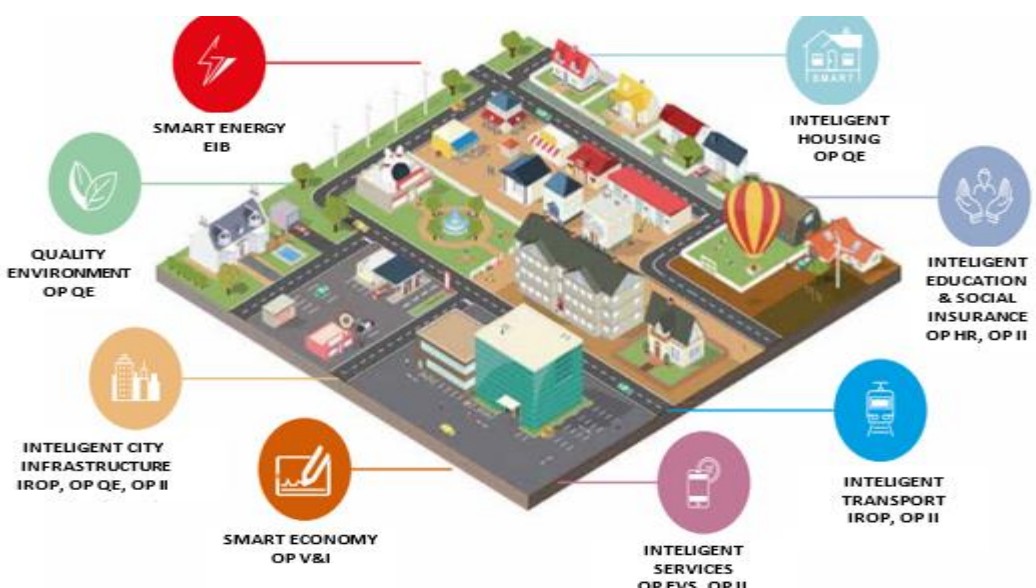

**Figure 1.** Smart solutions and funding scheme.

Only 70% of the surveyed SMEs perceive a positive benefit; therefore, it is also necessary to raise awareness of Smart Cities projects on both sides. The support has expanded to the creation of collaboration through co-working spaces, incubators and accelerator centres. The grant scheme financed by the energy giant VSE provides the financial and educational support for eastern Slovakia [49].

The National Project to Support the Development of the Creative Industry in Slovakia (financed from the European Regional Development Fund through the OPR and I scheme) [50] also supports innovative solutions and new start-up companies' foundation. Almost 1/2 of micro and 1/5 of SMEs used this support system to create solutions applied in various enterprise areas, or in energy and building management [47]. In this way, SMEs disposing of expertise can obtain the missing funding. Smart Cities projects are also focused on the issue of health, the sustainable environment of the population and healthy food, as well as digitization [38]. Despite the fact that there has been a shift in their development, the existing low awareness and the need not only to develop, but also to make available user-friendly solutions and many legislative and organizational changes constitute an obstacle to their development.

2.5.3. The Level of Digital Skills and the Digital Economy and Society Index (DESI)

If the EU states aim to implement the selected investment priorities financed from EU funds, they need to publish DESI index reports [51] as an overview of the conditions created for the development of the digital economy and society. This composite indicator is combined from 33 indicators that have been changed by the EC over the years. For this reason, target values are not set for the index, as was the case, for example, for the indicators of the Digital Agenda for Europe [52]. The Ministry of Finance of the Slovak Republic based its compilation on the national Strategic document for the growth of digital services and the infrastructure of the new generation access network (2014–2020) [53] and also on the Multiannual Financial Framework 2021–2027 [54].

The DESI 2022 indicator reports are mainly based on data obtained in 2021 in the area of digitization [51]. It determined that as a result of the COVID-19 pandemic, EU states have made progress in their digitization efforts, but are still struggling to close the gaps in digital skills, the digital transformation of SMEs and the deployment of advanced 5G networks. Slovakia ranked 23nd in DESI 2022. In its report, the EC notes that Slovakia is making progress, but it is not fast enough compared to other EU member states. The DESI key areas for the next periods were selected as follows:

- human capital,
- connectivity,
- integration of digital technologies,
- and digital public services.

As the area of the use of Internet services will no longer be monitored, an indicator on investments in ICT for environmental sustainability is added as an example. The DESI Country Reports also provide an overview of the digital investments and reforms included in the EU Member States' Recovery and Resilience Plans [32].

Comparing the results of the DESI index for the years 2018 and 2022, we can state that at least the basic level of digital skills of Slovaks reaches only 55% compared to 59% in 2018, which is slightly above the EU average (which also decreased from 57% in 2018 to 54%). Therefore, the conditions for digitization need to be created, as Slovak and other residents of disadvantaged groups are at risk of digital poverty, which makes them marginalized groups. The plan for the spread of the Internet until 2030 and the connectivity of households to ultra-fast broadband connectivity has shown progress, but Slovakia is below the EU average.

Slovakia's e-commerce score has also decreased: 13% of SMEs sell online compared to 17% in 2020 [51].

Due to the increase in threats in cyberspace due to the war in Ukraine, the Slovak government adopted the Ministry of Defence's "Action Plan for the Coordination of the Fight against Hybrid Threats" and online portal as a central hub for refugees from Ukraine and also an electronic form for temporary asylum registration.

As the number of users of electronic public administration among the Internet users has also decreased to 62% in opposition to the EU average value at 64%, the Slovak government aims to develop more digitally responsive public services and electronic healthcare by supporting the development of solutions that will be intuitive, easy to use and available in a mobile environment [54].

Most of the funds could be obtained from NextGenerationEU (EUR 723.8 billion), which will be spent through the Recovery and Resilience Facility (RRF). To receive support from the RRF, EU countries had to submit Recovery and Resilience Plans to the EC [55].

The European Declaration on Digital Rights and Principles (signed on 15 Decmber 2022) guarantees digital rights that imply e-connectivity for everyone, digitized education, trouble-free online access to public services, environmental safety and personal data protection. Other support is eInclusion, tied to monitoring specific outcome indicators (such as the percentage of the population with insufficient ICT skills for the labor market, etc.).

The COVID 19 pandemic is the basic factor that forced people to increase their own digital skills during the last two years. According to [38], in SR, like in EU, all generations were forced to expand their knowledge by self-study or self-education (40%) or using social contacts (in 37% of cases—with family, 32%—friends, 28%—colleagues at work) with the aim to control the use of e-communication, e-services or e-payments (e-health services (67%), portals (59%), to communicate with health centres (53%), and in 31% of cases to manage health insurance, connect with e-government, social and employment office, tax office or others). Nevertheless, according to the results of IT Slovak Association, research showed the decrease in the advanced digital skills (to 21% in DESI 22) that need to be improved [51].

The Digital Transformation Strategy of Slovakia 2030 [38] is a framework supra-ministerial government strategy that defines the policy and specific priorities of Slovakia in the context of innovative technologies and global mega-trends of the digital age (Artificial Intelligence, IoT, 5G technology) with the aim to develop a functioning information society and an innovative digital economy in which businesses can innovate and create sustainable jobs occupied by qualified and retrained workforce with advanced digital skills [51]. Even though in this regard, the Slovak Republic is on the tail of Europe [11], the Multiannual Financial Framework 2021–2027 (prepared by MIRRI SR) [54] does not primarily include education activities aimed at enhancement of digital skills of young people, even into



lifelong adult education. Therefore, more detailed knowledge of preferences of the groups will allow to better adapt the approach to ICT, as well as to develop their digital skills in a simple way.

The need to have equal access to the open internet, regardless of different status of citizens or different groups of society, different disadvantaged groups of citizens or different developed regions should be a matter of course. The goal is to make available to everyone the possibility to use the advantages of digitization [56].

To obtain information on whether the value of region increases ICT use, we proposed following hypothesis:

**Hypothesis 6:** *Increased willingness to obtain and provide information and trust in responsible authorities depends on an increase in population density.*

Only precise knowledge of the requirements and preferences of the population will support their interest in expanding knowledge potential and effective production of smart solutions. We will examine the influence of various factors (as well as the effects of the COVID-19 pandemic) on the level of awareness and use of e-connectivity in the following research.

### 3. Data Analyses and Methods

The main objective of the research was to determine to what extent selected independent (socio-demographic and socio-economic) variables influenced consumer behavior in order to predict changes in dependent variables (purchasing habits in the future).

We aimed to review the ways in which mandatory self-isolation during the COVID-19 pandemic has affected consumer behavior and health lifestyle and the role of ICT use. In order to conduct this review, we focused on examining the reason behind changing the way of buying and obtaining information. The following starting points were the basis for our empirical research activities, a basis for setting and clarification of various aspects to explore their impact on the perception and growth of the need for digitization:

- to describe the role of ICT use and consumer behavior changes,
- to examine the changes in consumer behaviors and their impact on changes in value perception and the ways of their increasing,
- to examine the ways to increase the willingness and need for participation in engaging in public dialogue.

Data sets were collected by the two surveys carried out by a questionnaire method collected in person, but also by e-mail. The representativeness of surveys consisting from the selected set of samples was verified using the Chi-square test according to the gender and age of the customers.

The first survey was realised in the period from December 2018 to April 2019. We researched the impact of various factors on the willingness to use ICT as a tool of electronic connectivity to facilitate purchases and increase self-awareness. The factors included solving security issues and revealing unfair practices in connection with the dual quality of goods. In this case, the return rate was 89.73%. The sample group consisted of 987 buyers who purchase products of daily consumption in the Slovak Republic aged 16 to 75, with approx. 85 respondents over 75 years of age, with the following gender distribution: women: 65.7%; men: 34.3%. The only important criterion was that they bought goods of daily consumption for themselves or their household.

In case of the second research, the return rate was 93.78%. This survey was implemented in the period from May 2022 to September 2022. The research sample consisted of a representative sample of 347 consumers from the Slovak Republic. The representativeness of the samples was ensured based on the age and income of respondents according to the Statistical Office of the Slovak Republic. Only consumers above the age of 18 were considered as people who contribute to the household budget regularly. The participants

were aged 18 to 66 and older, with the following gender distribution: women: 51.78%; men: 48.13%.

Based on the results of basic research, we determined that the changes in shopping habits depend on the types of various specific groups of Slovak consumers. Subsequently, in order to examine to what extent consumer behavior, general awareness and the degree of digitization depend on the selected factors, we have established several hypotheses. To verify them, the IBM SPSS Statistics 28 (New York, USA) software tools were used. First, a preliminary analysis of model variables was performed. Multiple regression analyses were used to assess statistically significant and statistically less significant variables and the strength of the relationships between them. In this way, it was possible to determine which of these variables has the strongest relationship with the dependent variable (participation or the need to clarify business transparency). We also attempted to compare the influence of socio-demographic or socio-economic factors on the preferences for methods of finding information and sharing knowledge about a given issue.

## 4. Results

Due to impact of COVID-19 pandemic, the new, previously insignificant factors started to significantly influence consumers. Factors such as the feeling of safety in the store, its location or accessibility from home became important for consumers during the pandemic. Mandatory self-isolation was a new phenomenon that consumers had not yet encountered in their lives. On the contrary, after the pandemic, the influence of personal reasons on shopping habits has decreased, but effects on the way of shopping still remain. What are the influences that permanently affect purchasing habits and what role do ICT capabilities play? We decided to obtain the answers to these questions by evaluating the theoretical knowledge and conclusions of our surveys. We verified respondents' opinions on the impact of the COVID-19 pandemic on a sample of 347 respondents, and evaluated the degree of willingness to use ICT to increase self-awareness about the subject of purchases on a sample of 987 respondents.

Changes in the purchased quantity

First, we researched the extent to which the lockdown affected the quantity of purchased goods.

Table 1 presents data on changes in quantities of purchased goods by consumers during the quarantine.

**Table 1.** Changes in the purchased quantity due to the mandatory self-isolation.

| Changes in the Quantity Purchased | Yes | No | Total |
|---|---|---|---|
| I shopped more | 12.79% | 16.85% | 14.86% |
| I shopped less | 51.16% | 51.74% | 51.43% |
| I bought the same amount as before the pandemic | 36.05% | 31.46% | 33.71% |

Data indicate that even if consumers were experiencing mandatory self-isolation, they did not significantly alter their shopping habits. It can be observed that more than half of respondents (51.74%) expressed that they did not buy less, some (36.05%) even asserting that they bought the same amount of goods as before the pandemic. Therefore, it is possible to conclude that this measure implemented to stop the spread of the virus had no major effect on shopping habits. Based on this, it can be assumed that despite the restrictions, customers quickly adapted to these changes and sought for new ways of shopping.

As more than half of respondents stated that they do not buy less, with one third even buying the same amount as before the pandemic, the first part of Hypothesis 1 was not confirmed as true. This means that the quantity of purchased products does not depend on the decrease in income.

We further investigated the factors (properties of products or the ways of buying) that significantly influenced consumers' habits.

Since the need for self-isolation persisted during the pandemic, the limiting factor was the unavailability of stores or products.

Table 2 shows the measures of significant dependencies between consumer behaviour preferences and selected socio-demographics and socio-economic criteria and product attributes.

**Table 2.** Significant interdependencies of selected criteria and consumer behavior preferences.

| Attributes | Gender | Age | Level of Education | Income |
|---|---|---|---|---|
| Price | | −0.082 ** | | |
| Quality | 0.116 ** | −0.065 * | 0.136 ** | 0.119 ** |
| Health risk | 0.071 * | | | |
| Product availability | | 0.065 * | | −0.068 * |
| Store availability | | 0.079 * | | −0.197 ** |
| Experience with the product | | | 0.115 ** | |

\* $p < 0.05$, \*\* $p < 0.01$.

As can be seen from Table 2, independent variables such as the criterion of unavailability of a store or product significantly influenced the depended variable of consumer behavior only in the case of selected socio-demographic and socio-economic criteria such as age and income. We determined that there exist significant interdependences between preferences for criteria such as store or product availability depending on income and age categories.

In the case of age groups, customers were limited by the unavailability of the store (Phi = 0.079), and then the unavailability of products (Phi = 0.065), both at $p < 0.05$; the dependence is slightly directly linearly dependent. Regarding gender, both criteria affect men more than they do women. Low income groups are mostly limited by the increased store unavailability (Phi = −0.197 at $p < 0.01$) as well as the product unavailability (Phi = −0.068 at $p < 0.05$).

Income

As we further determined by observing the data, the obstacle in the form of the inaccessibility of the shops is the most limiting for lower income groups and older age groups. In case of unavailability of the product, the dependence is changed in an inversely proportional manner with the income. Furthermore, there were differences in the factor based on consumers' segmentation by household income and shopping habits (Table 3).

**Table 3.** Results showing whether mandatory self-isolation changed consumer shopping habits depending on household income.

| Household Income | Yes | No |
|---|---|---|
| 400 euro or less | 85.71% | 14.29% |
| 401–1000 euro | 48.15% | 51.85% |
| 1001–1600 euro | 54.41% | 45.59% |
| 1601 euro or more | 36.96% | 63.04% |
| Total | 49.14% | 50.86% |

According to the data, respondents with the lowest income were more likely to alter their consumer behavior during the mandatory self-isolation. On the other hand, 63.04% of consumers with the highest income did not implement any changes during quarantine. Using Pearson correlation coefficient, the medium strong indirect dependency

(Phi = −0.450, *p* < 0.01) was discovered between these two variables confirming the finding that income and changes during self-isolation are related.

This confirms the validity of the second part of Hypothesis 1, meaning that changes in purchasing habits depend on the income decreasing in the lowest income groups.

Quality and health risk

Another criterion for which significant mutual correlations of dependencies were calculated was quality. According to Table 2, changes in quality of products slightly influenced all categories; however, the most interdependence exists among more educated people.

Interest in quality increases with the increase in education level of consumers (Phi = 0.136, at *p* < 0.01); dependency is moderate.

Women are more interested in quality (Phi = 0.116, at *p* < 0.01) and health risk factors (Phi = 0.071, at *p* < 0.05). In the case of age, younger consumers are more interested in quality, and with the increase in age, the interest decreases.

Interest in health risk increases with age, but no statistically significant dependence was observed.

Overall, consumer behavior is most affected by high price and low quality (both confirmed with mean = 0.79), where quality is slightly more preferred (st. dev. = 0.405) as opposed to price (where st. dev. = 0.407).

Since there exists a mutual slightly positive interdependence between income categories and quality (Phi = 0.119, at *p* < 0.001), we accept the validity of Hypothesis 2 as true. We can state that the changes in the interest in quality and health risk depend on the income.

Low-income groups are most at risk of purchasing low-quality of cheap products when they attempt to save either on food of lower quality or that of worse composition.

Next, we investigated the product features that may be dangerous to health and are unsuitable for consumption, which and may significantly change the consumers' purchase decisions. Therefore, we also investigated the ways in which changes in consumer preferences depend on the awareness about the composition of goods.

Table 4 shows mutual dependencies of selected criteria.

**Table 4.** Mutual correlation table of criteria dependency.

| Attributes | Substitute Quality | Raw Material Ratio | Component Ratios | Poor Quality of Processing | Product Quality |
|---|---|---|---|---|---|
| **Substitute quality** | | 0.122 ** | | | |
| **Raw material ratio** | | | | | |
| **Component ratios** | −0.189 ** | 0.109 ** | | −0.113 ** | |
| **Poor quality of processing** | −0.085 ** | −0.229 ** | −0.113 ** | | |
| **Product quality** | −0.216 ** | −0.18 ** | −0.079 * | | |
| **Age** | 0.068 ** | | −0.076 ** | 0.083 ** | |
| **Income** | | | 0.077 ** | | 0.074 * |

* *p* < 0.05, ** *p* < 0.01.

Considering the substance of the problem of dual quality, respondents identified it as a significant ethical–legislative problem (46.15%) with slightly statistically significant relationship (where Phi = 0.022). If it is an economic problem (19.13%), then it is only an ethical problem (14.05%). We detected a moderate indirect dependence between ethical–legislative and ethical nature (Phi = −0.412).

A difference in opinions was observed according to age categories towards income categories with a weak dependence Phi = −0.213, which is slightly significant (in a different millennial group aged 26–45 years).

Media and willingness to participate in problem solving

Quality can be considered a driving force that develops the need to be informed personally, but also through the media. However, there also exists the threat of unfair practices which arises from the already solved problem related to the dual quality of food.

According to our statistical evaluation of answers to the question of dual quality, we can state that more than 89% of respondents are aware of the problem of dual-quality of foods (mean = 0.89 with st. dev. = 0.31). People in productive age (from 26 to 45, a total of 395 respondents) or people in older productive age (from 46 to 65, a total of 278 respondents) are more interested in the issue of dual quality products than others (Phi = 0.212 at $p < 0.01$). Regarding the question of whether the respondents would be willing to inform themselves about the problem of dual quality of goods, 82.37% would be willing to do so (mean = 0.83, st. dev. = 0.38), especially secondary school educated and university educated people and those who make purchases. Others would not be willing to inform themselves at all. Women (mean = 0.91, st. dev. = 0.72) are more willing to do so than men (mean = 0.82, st. dev. = 0.77).

On the other hand, the millennial generation informs themselves less than the people of the older productive age, but those that do assign it a higher importance.

In Table 5, we summarized the tendency of the willingness to stay informed about the issue of products dual quality depending on the growth of the awareness.

**Table 5.** Willingness to stay informed about the issue of dual quality.

| According to: | Willingness to Stay Informed |
| :---: | :---: |
| **Gender** | Women prefer it more than men do |
| **Age** | Rises/is decreasing |
| **Degree of education** | Rises |
| **Income** | Rises |

We determined that the perception of the need for participation in solving the problem generally does not depend on the level of knowledge of the problem being solved. We infer this from the fact that there are various groups that do not like to participate, even as their awareness increases.

Hypothesis 3 was only partially confirmed as true due to the different level of willingness to participate in the problem solving regardless of the increasing level of awareness. It is possible that the lower level of willingness to inform others and to participate in solving quality issues was caused by the lower level of digital skills (due to the necessary eID notification for portal access to a specialized website). We aim to address this question in the future.

Furthermore, we investigated the other criteria that depend on the change in the purchasing behavior of the respondents. According to our findings, consumer decision-making is mostly based on the level of experience with the product influenced also by the level of information.

Table 6 shows measures of significant correlation dependencies between dependent variable of level of awareness and selected independent socio-demographics and socio-economic criteria of selected type of goods.

We can state that changes in the level of awareness depend mostly on consumer gender, income and on the level of education. The level of awareness depends on the various types of criteria and digital skills of consumers. Young people, women more so than men, higher income groups and people with higher education are the most experienced with cosmetics. Women prefer to buy more detergents and products such as Cosmetics, Sweets, Cleaning products or Foods.

In addition, we investigated which consumer experiences with one product's attributes relative to the other were more likely to influence their dissatisfaction, changing their purchasing habits.

**Table 6.** Significant dependencies between selected criteria and level of awareness.

| Attributes | Gender | Age | Level of Education | Income |
|---|---|---|---|---|
| **Foods** | 0.072 * | 0.123 ** | | |
| **Sweets** | 0.090 ** | | | 0.067 * |
| **Cosmetics** | 0.132 ** | | 0.127 ** | 0.066 * |
| **Cleaning products** | 0.084 ** | | 0.068 * | |
| **Detergents** | 0.153 ** | 0.08 * | 0.064 * | |
| **Clothes** | | | | 0.068 * |
| **Alcoholic and non-alcoholic drinks** | | | | 0.088 ** |

\* $p < 0.05$, \*\* $p < 0.01$.

We evaluated mutual dependencies of the criteria on each other. We found out that if there exist any significant relationships between the criteria (at $p < 0.01$ or $p < 0.05$), these relationships are always non-linear, that means an indirect increase in consumer dissatisfaction. Computed values are shown in Table 7.

**Table 7.** Values of mutual correlation dependencies between selected criteria.

| Attributes | Price | Quality | Health Risk | Product Availability | Store Availability | Experience with the Product |
|---|---|---|---|---|---|---|
| **Price** | | | −0.189 ** | −0.085 ** | | |
| **Quality** | | | | −0.194 ** | −0.200 ** | |
| **Health risk** | −0.189 ** | | | −0.113 ** | −0.079 * | −0.297 ** |
| **Product availability** | −0.085 ** | −0.194 ** | −0.113 ** | | | −0.225 ** |
| **Store availability** | | −0.200 ** | −0.079 * | | | −0.197 ** |
| **Experience with the product** | | | −0.297 ** | −0.225 ** | −0.197 ** | |

\* $p < 0.05$, \*\* $p < 0.01$.

We can deduce that if the products are less available, the consumer fears that products will be of poor quality, endanger their health or be too expensive. Then, the consumer's shopping habits mostly depend on their own experiences. If access to the store is limited or not allowed, the consumer is concerned about the quality of the products or about lack of experience with other similar products. The lower the amount of personal experience with the product, the higher the stress rate of consumer regrading health risks (Phi = −0.297). Measures of the government and regional institutions and easy way of e-participation problem solving will play an important role here.

Next, we investigated the factors influencing the respondents' purchases and the respondents' preferred sources of information.

As we can deduce, changes in shopping habits significantly depend on respondents´ gender, age and education level. The respondents surveyed consider their own experience to be the most credible sources of information. The respondents rely on their own experience, eventually experience of their family members or friends. There is a negative linear dependency between age and family and also positive significant dependency between gender and level of education.

Table 8 shows sources of information that are the most reliable for the respondents.

In the case of family, there exists a slightly higher nonlinear age dependency.

In the case of media, we can state that there exists only mutual dependency between media and age (Phi = −0.101). Community is a more important source of information in criteria of age, level of education and income.

**Table 8.** Significant correlation dependencies on information sources.

| Attributes | Gender | Age | Level of Education | Income |
|---|---|---|---|---|
| **Own experience** | 0.074 * | −0.076 * | 0.07 * | |
| **Family** | | −0.128 ** | | |
| **Acquaintances** | | −0.065 * | | |
| **Community** | | 0.103 ** | −0.095 ** | −0.08 * |
| **Media** | −0.101 ** | | | |
| **Results of scientific research** | | | | |

\* $p < 0.05$, \*\* $p < 0.01$.

Furthermore, we considered the changes in shopping habits in correlation with the changes in type of information according to age groups. In this case, the most preferable criterion is family experiences, with own experiences being most often preferred by up to 32.68% of respondents with income from EUR 801 to EUR 1000 and from EUR 1001 to EUR 1200; these groups do not prefer to rely on their acquaintances' knowledge. The consumers in the age category from 26 to 45 years tend to trust their own experiences (Phi = 0.132 at $p < 0.01$). Media preference is also age-dependent (Phi = 0.107 at $p = 0.011 < 0.05$), young people, unlike older people, prefer media.

In the case of gender, there exists a slightly significant dependency. Shopping habits change slightly significantly depending on the ways of obtaining information according to consumers' gender. Men use media to be well informed more often (Phi = −0.101 at $p < 0.01$) than women, who instead rely on their own experiences (Phi = 0.074 at $p = 0.05$).

Therefore, Hypothesis 4 was confirmed as true. It means that changes in shopping habits are slightly significant depending more on changes in one's own experiences as well as community than on media use in various types of socio-demographic criteria.

Since women predominantly shop for their families, they are more interested in leading a healthy lifestyle and are willing to change their habits. Lockdown forced them to change their shopping habits. Because of this, we compared the changes in preferences between the independent variable of gender and the dependent variable of shopping habits due to mandatory self-isolation.

Table 9 shows the data on the effect of the mandatory self-isolation on shopping habits based on gender.

**Table 9.** Results showing whether mandatory self-isolation changed consumer shopping habits.

| Gender | Yes | No |
|---|---|---|
| **female** | 55.93% | 44.07% |
| **male** | 35.09% | 64.91% |
| **Total** | 49.14% | 50.86% |

It can be observed that male consumers were significantly less likely to change their habits during the self-isolation (35.09%) than female consumers (55.93%).

More than half of women stated that self-isolation changed their shopping habits, and more than 64% of men were less likely to change their habits.

Therefore, we considered the mutual correlation between distribution of household income between consumption and savings and the gender of the consumers in our sample. Mutual correlation (Phi = 0.138, at $p = 0.01 < 0.05$) was slight, but significant. We can claim

it with 90% probability. This means that shopping habits depended on the effort of women and low-income groups to divide expenses efficiently.

We can conclude that Hypothesis 5 is confirmed as true. Women are more willing to change their shopping habits than men. The reason is the need to spend money more efficiently.

The ways of awareness and transparency of value chains

ICT technologies and online shopping possibilities played an important role. We decided to assess the effect of the use of ICT tools on the changes of shopping habits.

We have detected a linear dependence (Phi = 0.313 at p < 0.01) of the gender of the person responsible for shopping in the household and its source of information. We can also state that people who make purchases tend to want to be more informed (Phi = 0.132 at $p < 0.01$) compared to those who do not.

As a source of information, those making purchases would prefer the following options: media coverage in dailies or weeklies (mean = 0.47 with st. dev. = 0.499), a transnational internet portal (page) (mean = 0.20 with st. dev. = 0.399), together with Phi = −0.279. Only approximately 14.83% of respondents would prefer the current information provided by the State veterinary institutions or Slovak Trade Inspection Authority, 14.08% would like information to be provided in the form of regular reports by the Prime Minister and also the Ministry of Agriculture and Rural Development of the Slovak Republic (Phi = 0.304). Only 10.73% of respondents would welcome a regular report from the European Commission.

The trust in government measures increases slightly with the age of the respondents.

In regard to the tool with which they would like to inform themselves, respondents most often choose a web portal (after registration, it is possible to directly insert information, photos, etc.) (mean = 0.34, st. dev. = 0.475), sharing via social networks on a specialized profile, with Phi = at the level of −0.240, (mean = 0.22, st. dev. = 0.41), and via the mobile application (mean = 0.14, st. dev. = 0.345). They also trust the state veterinary institutions or Slovak Trade Inspection Authority (mean = 0.17, st. dev. = 0.377). Therefore, the respondents are open to new technologies, but they prefer easy access to these tools. Unavailable user-friendly applications and login methods tend to be a problem.

We also investigated the willingness to obtain and provide information in order to increase self-awareness and trust in responsible authorities with regard to the increase in population density.

An information about unfair practices is provided by the specialized website. This source is most preferred by residents of areas up to 10,000 inhabitants and up to 5000 inhabitants, followed by respondents from larger settlements. In small settlements, the situation is almost the same. This means that the size of the settlement does not significantly affect one's willingness to use different forms of electronic communication. The lowest income groups prefer social networks (Phi = 0.062) or a specialized portal, or they trust in state control institution, followed by mobile applications.

Hypothesis 6 was not confirmed as true. There are no differences in the use of different types of electronic communication tools by different population groups to increase general awareness.

Sustainable development should be obtained in an understanding and friendly approach to visitors in harmony with nature and with regard to the support of the local population.

## 5. Discussion

During the COVID-19 pandemic, consumers began to change the structure of the products they buy. Up to 13% of Slovak consumers (out of 347 respondents) stated they bought more products than they did before the pandemic. Only a small part of consumers stated that they would have to buy the products due to their unavailability (32%). In addition, other results of the survey that we conducted on representative samples of Slovak consumers resulted in interesting findings that can be used to estimate future trends in the development of consumer behavior.

Next, the issue lies in the finding that young people or people with low or no income do not consider saving money. It confirms the findings that correlation between distribution of household income (between consumption and savings) and the age of the consumers is slightly positive, where Phi = 0.138 at $p < 0.05$. This also confirms the finding that the older the consumer, the better the proportion of consumption [18]. We found out that mutual dependency of consumers income and their shopping habits is moderately strong (Phi = $-0.450$, $p < 0.01$).

In contrast to this, the concern for consumers' own health has increased. We determined that consumers with the lowest income were more likely to alter their consumer behavior and buy cheaper products, always with lesser quality. On the other hand, approx. 63.04% of consumers with the highest income did not change their shopping habits. Hypothesis 1 was not confirmed as true.

Low-income groups are most at risk of purchasing low-quality or cheap products when they attempt to budget, either for food of lower quality or worse composition, which can lead to health damage, for example, due to allergies [57].

As shown in the Table 4, consumers are less disgusted by the poor quality of product processing if the raw material ratio, product quality or component ratio of the substitute product is higher. Although the "New Deal for Consumers" directive [27] strengthens the powers of consumer protection authorities at both legislative and ethical levels, the media handling of such cases is important. The correlation coefficient Phi = $-0.240$ indicates a slightly indirect correlation between the age of the respondents and the preferred type of media—sharing information (on social networks on a special website). Consumer protection is also enshrined in The Digital Transformation Strategy of Slovakia 2030 [38], where the goal is to create tools and conditions for the protection of all population groups against the negative impact and abuse of ICT and against the deepening of the generation gap in the use of digital technologies.

Since there exists a mutual slightly positive interdependence between income categories and quality (Phi = 0.119, at $p < 0.001$), we accept the validity of Hypothesis 2. We can state that the changes in the interest in quality and health risk depend on the income amount.

Changes in consumer behavior were initiated due to decrease in store or product availability. We determined that the obstacle in the form of the inaccessibility of shops is the most limiting factor for older age and low-income groups. Unavailability of the product (Phi = $-0.225$) or the store (Phi = $-0.197$) increase interest in experience with the product obtained personally, with the media use also increasing due to information search in connection with interest in its quality, level of health protection or price at all (attempting to save more when investing or buying). An increase in the influence of Internet use was also confirmed on changes in shopping habits due to the need for professional information searching (an increase by more than 50% compared to the period before the COVID-19 pandemic).

A significant increase occurred especially among women. It was stated that mandatory self-isolation changed their shopping habits more (55.93%) than in case of men, where approx. 64.1% of men stated they have not changed their shopping habits. Women are also adapting, and since they shop for their families, they are more interested in leading a healthy lifestyle and are willing to change their habits. Hypothesis 5 was confirmed as true. As mutual correlation between distribution of household income between consumption and savings and the gender of the consumers exists, we can confirm that women are more willing change their shopping habits than men. The reason is concern for savings in accordance with quality and minimization of health risks.

We determined that the perception of the need for participation in solving the problem generally does not depend on the level of knowledge of the problem being solved. We infer this from the fact that there are various groups that do not like to participate in decision-making, even as their awareness increases.

Hypothesis 3 was only partially confirmed as true due to the different level of willingness to participate in the problem solving regardless of the increasing level of awareness. It is possible that the lower level of willingness to inform others and to participate in solving quality issues was caused by the lower level of digital skills (due to the necessary eID notification for portal access to a specialized website). Our research had certain limitations because we did not examine the degree of willingness to use such tools in accordance with all possible influences and taking into account socio-demographic or socio-economic criteria. We aim to address this question in the future.

Furthermore, we explored the reaction of the government and regional institutions to these conditions and also the conditions for the development of e-connectivity. The financial crisis increases the risk of unfair practices. It is more than necessary to conduct an overview of these practices, since consumers' protection is essential. In the context of food safety in the EU, there is a Rapid Alert System (RAS) focusing on, for example, the presence of foreign objects in food or the concentration of food supplements [24]. The international standard ISO 22000: 2018, valid after 29.06.2021 [30], guarantees the basic management tool for monitoring compliance and gaining customer confidence as the proof of the introduction of a food safety management system by producers.

Electronic support for publishing and receiving information on product quality issues and unfair practices is already available in Slovakia. In addition, the Slovak government intends to support online electronic healthcare as a part of its recovery plan. The problem is, consumers rarely use such ways of communication. The low level of preference for these forms of communication was also confirmed by the results of our research.

It is assumed that the level of knowledge about product quality issues and unfair practices will increase the willingness to participate. In addition, Slovak government intends to support online electronic healthcare as a part of its Recovery plan [55]. Online Health Services and their acceptance, especially among the older generation, requires the need to build an appropriate relationship with them. As was determined, the family or communities play an important role of an advisory body here, providing support for all online health-related activities and many other matters as well.

Therefore, Hypothesis 4 was confirmed as true. This means that slightly significant changes in shopping habits depend more on changes in one's own experiences and also community than on media use and its preferences in various types of socio-demographic criteria.

Considering the fact that the pandemic has already been followed by other adverse events in the regions, the growing inflation will make the already poor financial situation even worse. The worst situation is observed in marginalized groups such as the older generation, the unemployed or low-income families, which do not possess broad, wide connectivity or whose income is very low or almost non-existent. In addition, lower interest in ICT was confirmed, which can be caused due to the lack of understanding the need to adapt to the digital age, the lack of money or knowledge of the ways to use e-services. Subsequently, it is necessary to examine which environment is most suitable as a communication channel in marginalized groups, as well as ways to obtain financial resources or to improve the quality of online activities.

The need to improve basic skills as well as implement advanced digital skills into society is growing with the growing importance of media; the results of a survey (Turkey [5]) revealed that 54% of teenagers relied on social media and 50% on YouTube for obtaining various types of information.

Finally, we conclude that changes in shopping habits depending on wider digital skills of inhabitants as skilled consumers do not primarily depend on the size or location of the region and its degree of digitization, which was also confirmed by the verification of Hypothesis 6 of our research. Rather, these changes depend on previous users´ experiences and their digital skills, and also user-friendly applications available to everyone. An above-average share of digitally illiterate people can still be seen in the less-educated, less-qualified, economically inactive (retired, unemployed), socially weaker and rural parts

of the population (villages with up to 5000 inhabitants) [38]. The problem of a way to adapt the environment and forms of education to the needs of users persists.

This knowledge is more than necessary, and therefore we aim to devote ourselves to researching this issue in the future.

Virtual shopping poses a significant challenge to those aiming to enhance support of digital economics. It provides an opportunity to obtain an electronic database of contacts, to keep customer preferences and to speed up business relationships and also personalize offers according to customer needs. Such trading supports balanced modes of customer contact and decision making.

AI also brings its benefits. As head representatives of KPMG's in [19] stated, 53% of retail workers state that the adoption of AI accelerated due to the need to respond to the pandemic. Subsequently 69% of those who deployed new technologies stated that the rate of added value was much higher than expected. An example is the possibility of reporting a telephone line outage or faster provision of services and their improvement to customers in case of overload of information lines. The question is whether such services are acceptable for the customer's comfort and satisfaction due to missing human contact by phone. In any case, it increases the importance and need for innovations and makes them more widely understood by society.

Digital skills enhancement

Support can be developed through activities aimed at mutual information sharing, learning and acquaintance, which will strengthen the perception of the roles of various institutions and public organizations. An example of ways to improve the digital user skills of marginal groups, for example, seniors or other population groups [58], is training courses financed under the Slovak Recovery and Resilience Plan [52].

In the first phase, Slovak seniors threatened by digital poverty and low connectivity learn to control technical devices and software, but also operate safely in the online space. Teachers are learning what procedures to choose in order to increase the digital skills of other groups of seniors as effectively as possible. The participants will maintain the skill by receiving a tablet with the accessories they worked with during the lesson, and with starter data pack pre-purchased. This is the way to complement uniform environment for knowledge sharing, This is the way to complement uniform environment for knowledge sharing, increasing the possibilities for successful digital user skill enhancement. It facilitates them to use online ordering and to communicate with other seniors via social networks. It helps participants use online ordering and to communicate with other seniors via social media. Doctors can communicate with a patient online, increase awareness and also provide medical education to the public online (e.g., LCS KNIS—Ordering patients (fnsppresov.sk)).

Barriers in the integration of digital technologies are the ongoing administrative burden, low awareness of financial possibilities and financial instruments where support is hindered by challenges and related types of training [59]. As part of the publication of the Berlin Declaration [60] (2020) as well as the Digital Compass 2030 [61], conditions were formulated to support the development of digitization in EU countries.

## 6. Conclusions

The effects of the COVID-19 pandemic, i.e., inflation, as well as the armed conflict have led to significant changes in market conditions throughout the world. The consequences of the pandemic have been borne by individual national economies and individual businesses even in 2022. One of the consequences of the pandemic is also major changes in consumer behavior and purchasing habits.

There are certain disproportions that lead to different considerations and the need to further investigate and assess the influences of various factors. When we examined whether the mandatory self-isolation caused a change in shopping habits, overall, the respondents stated that their habits did not change significantly. Men did not significantly alter their shopping habits. Women stated that there were changes for them (55.93%) rather than for men (35.09%). Changes in shopping habits during mandatory self-isolation depend on

gender. This may be related to the greater preferences of women in the area for product quality and care for health protection in the family. Women are more motivated to use e-connectivity to enhance their own awareness, and also to use e-commerce as a basic environment for making purchases.

In regard to increasing prices, significant dependency was recorded among young people who have no or lower income. They feared the increase in prices and decrease in quality of products. Our recommendation is to spend only on the essential goods. As they possess digital skills, they can examine the quality of products of daily consumption more carefully and devote more attention to issues of consumer trust, especially in connection to regional producers and sellers. In this way, the role of social media grows.

In order to attract families with lower income, new forms of vacation activities are offered reducing the provider cost at the same time, result in the need to attract customers even at a price of at least the median cost of vacation. We propose to personalize the offer, focusing more on families with children and their needs. In order to restore the gastronomic tradition, it is important for entrepreneurs to emphasise the naming of their business associated with regional cuisine and then offer a local product with selected dishes from local cuisine and homemade products [20]. Consumers in the USA and in Slovakia are also driven by economic reasons to support small retailers, the local community and the locally manufactured products. Due to various pandemic restrictions, purchasing of regional products was popularised, but as costs of such products are rising at the same time, support is needed. The opinion of the public varies, as 13% of respondents express more trust in local shops while 15% of respondents mistrust local shops being convinced they are not as supervised from the point of view of the quality of goods as large chains. It is also due to a lack of product availability and higher prices, which was also confirmed in Poland and Turkey [5]. However, local products pose less health risks, including the lower risk of other environmental impacts on the customer's health. This forms the basis for healthy home eating. The factor that most differentiates changes in consumer behavior is geographical, although consumer shopping habits are different from country to country. Turkish students have a tendency towards panic buying, stockpiling of food, and expressing concern about losing their source of income. Polish millennials state that despite limitations on mobility, their physical shopping was still the most popular form of purchasing (51.3%) according to the size of regions or reduced deliveries of food products. In Slovakia, the situation is similar.

Furthermore, we determined out that even if the government creates conditions for the support and involvement of regional producers (who have innovative potential in contrast to employees of public organizations), concerns are expressed about the effectiveness of innovative solutions (70% of SMEs). The risky nature of such solutions results from obstacles due to the need to simplify access to services in the mobile environment for everyone, regardless of their level of digital literacy. Innovations are needed, but they are difficult to finance.

In order to obtain financial support within the Mechanism for the Support of Recovery and Resilience, we recommend SMEs to obtain a voucher as a way to develop their research and innovation activities. In this way, SMEs can enhance mutual cooperation and knowledge transfer by building digitization and innovation ecosystems based on mutual relationships. There are vouchers for innovation, digitization and also patent vouchers as a support for start-ups when expanding operations at the EU level and worldwide [62]. As the EU declares, the EC will support the deployment and use of notified eIDs also in the private sector, also helping to enhance innovation in specific regions in Europe. This is the way to enhance digitalization in regional development, which allows to increase people's interest in accessing digital technologies and consequently their willingness to participate in solving issues pertaining to regional life [63].

We recommend to explore mutual dialogue with the aim to assess the need for legislative amendments and other changes with regard to local conditions. Establishment of regional associations is a way to wider cooperation and enhancement of smart solution

specification [59]. An example is Slovakia's own and global awareness of dual quality in the area of food quality control, where, based on the suggestions of customers, authorities at the national level, as well as the EC, which were trusted by the respondents of our survey (25.56% in total), supported steps to solve this problem. Media coverage led to an increase in awareness, which led to the adoption of sets of measures and changes to directives in the EU and in the countries themselves [37].

Since customers prefer a visual environment, we also recommend that sellers create personalized advertisements on social networks where they can use the visualization of the value chain, from the cultivation of products to the preparation of meals and the offer of ready-made meals or events with the recommendation of well-known personalities or influencers.

As the difference in the quality of goods in the EU significantly affects consumers (81.1%), we recommend enterprises to increase transparency of their production processes. A way to control healthy goods is to supervise the entire value chain. The blockchain technology offers an opportunity to streamline their value chains. We recommend that enterprises create and share a database of real data, where consumers can follow the entire production chain from the reception of raw materials (e.g., buying milk from the farm) to the processing of the finished product [64]. The use of this technology requires expanding the functioning system, but also users' skills and knowledge enhancement for those with advanced or expert skills.

Despite that reducing health risk remained the priority, nowadays, people buy cheaper, albeit lower quality food. The consequences of saving on quality food can be manifested in increased morbidity, while an increase in the threat to physical and mental performance can cause increased costs for employers and the state. Approximately 73% of respondents prefer homemade meals. The allowance for food is insufficient for 35.5% of respondents. We infer that employers know the ways to influence the quality of food, and therefore also the quality of life, but the entire solution to the problem of rising food prices should not only depend on their decision [21].

Finally, we can conclude that the pandemic and other influences have caused an increase in digital skills, but it is not a consequence of the pandemic. The level of using online purchasing strongly depends on whether consumers have used the products they are purchasing before. This is also confirmed by the slight growth in basic skills (up to 55%) but the decline in advanced digital skills (21%) [61]. People want to search for highly specialized information that they cannot find personally, for example, when making a purchase. Those who have experience with the e-environment from the pandemic period stated that they would continue to purchase goods or foods online in future (approximately 35%). There is a reason for changing the purchasing behavior: the higher use of online purchasing is related to the convenience of shopping at home, the need to save, to buy products of the same quality and receive a cheaper delivery of goods, but mainly to the need to protect one's health in the long term.

In our next research, we want to find out how changes in digital skills and ways of developing them can affect the shopping habits of consumers, especially in marginalized groups. Furthermore, we focus on defining ways to develop their acquisition, as well as what smart solutions can make shopping more comfortable.

**Author Contributions:** Conceptualization, M.P. and L.V.; Methodology, M.P. and L.V.; Software, M.P.; Investigation, M.P.; Resources, M.P. and L.V.; Supervision, M.P.; Funding acquisition, L.V. All authors have read and agreed to the published version of the manuscript.

**Funding:** This research received no external funding.

**Informed Consent Statement:** Not applicable.

**Data Availability Statement:** All data has been present in main text.

**Acknowledgments:** This contribution was published within the project No. 1/0134/22, "Changes in consumer behavior due to the COVID-19 pandemic with intent to predict its development".

**Conflicts of Interest:** The authors declare no conflict of interest.

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
