# Peer review of "The Impact of COVID-19 on Purchase Behavior Changes in Smart Regions"

_computers, doi:10.3390/computers12020038_

Round 1
Reviewer 1 Report
Tables 5,6,7 have row ( column) Total and it is always 100%, this is not needed for presentation
Author Response
Tables 5,6,7 have row ( column) Total and it is always 100%, this is not needed for presentation.
- rows were deleted
Reviewer 2 Report
This paper is of sound quality on a subject deserving the Journal's attention. This study deals with consumer behavior due to different degrees of digitization and innovations as the driving force of smart regions after The Covid-19 pandemic. Employing regression and correlation analyzes, this paper identified barriers that defend sustainable development, and also to wider public dialogue and cause unfair practices that worsen consumers´ health and their quality of life. Overall, the paper is well written and well structured, therefore it is easy to follow and builds a clear conclusion from the data. Generally well written but requires some editing and revision.
1) In literature review: this study well reviewed prior researches. however, Background and Theoretical Context would be a part of literature reviews. The author requires reconsidering section structure, particularly the literature review part.
2) hypothesis would be developed based on literature reviews (in the case of empirical study) or the events (in the case of ground theory). However, this study displayed the hypotheses at the Methodology part.
In addition, to develop hypothesis, this study is required further literature reviews to present theoretical supports for each sub-hypothesis.
2) the Methodology part would be updated to present Research design, data analysis method, and processes.
In table 1: red value (required checking)
The processes for data analysis are appropriate and the results of it are clearly described. However, this paper just described the results of data analysis. To improve the quality of this study, author(s) need to extract more clear implications in both theoretical and practical perspectives as a discussion of the results. Additional explanations are required to link the results of data analysis and conclusions. Research conclusion (practical implication) part is weak, focusing on data analysis (enumerate bits of information). Additional explanations incorporating theoretical and practical are required.
The quality of communication is appropriate. however, in order to clearly present research objectives, I recommend putting the summary section in results and discussion part. Generally well investigated but requires some editing and revision.
This paper is of sound quality on a subject deserving the Journal's attention. Overall, the paper is easy to follow and builds a clear conclusion from the data, but requires some editing and revision.
Author Response
Rev 2
- In literature review: this study well reviewed prior researches. however, Background and Theoretical Context would be a part of literature reviews. The author requires reconsidering section structure, particularly the literature review part.
- the contribution structure has been revised
2) hypothesis would be developed based on literature reviews (in the case of empirical study) or the events (in the case of ground theory). However, this study displayed the hypotheses at the Methodology part.
- Hypotheses were reformulated and another literature review was added
In addition, to develop hypothesis, this study is required further literature reviews to present theoretical supports for each sub-hypothesis.
2) the Methodology part would be updated to present Research design, data analysis method, and processes.
- The structure of the contribution was revised according to comments
In table 1: red value (required checking)
- the table was modified, the value presented an independent correlation
The processes for data analysis are appropriate and the results of it are clearly described. However, this paper just described the results of data analysis. To improve the quality of this study, author(s) need to extract more clear implications in both theoretical and practical perspectives as a discussion of the results. Additional explanations are required to link the results of data analysis and conclusions. Research conclusion (practical implication) part is weak, focusing on data analysis (enumerate bits of information). Additional explanations incorporating theoretical and practical are required.
- in the revision, the theoretical analysis was reworked and also from a practical point of view, the discussion was supplemented
The quality of communication is appropriate. however, in order to clearly present research objectives, I recommend putting the summary section in results and discussion part. Generally well investigated but requires some editing and revision.
- editing and revision was made according to comments with the addition of a discussion of the problem and also an evaluation of the results and recommendations
This paper is of sound quality on a subject deserving the Journal's attention. Overall, the paper is easy to follow and builds a clear conclusion from the data, but requires some editing and revision.
- the revision was carried out and the contribution was reworked
Reviewer 3 Report
This study tried to compare the degree and nature of the interdependencies varied by influence of sociodemographic or socioeconomic factors on shopping habits and their quality. However, it needs to be improved in many ways. Please check the comments as below.
1. Is 'good safety' a typo for 'food safety' on Line 248?
2. The operational definition of the variable has not been presented. Also, since the survey item is not presented, it is impossible to determine the suitability of the results of this survey.
3. In the result of Part 5, the adoption and rejection of hypotheses are presented several times, which is very confusing. Please reorganize it in general to increase readability.
4. Conclusion part should separate the implication and limitation.
Author Response
- Is 'good safety' a typo for 'food safety' on Line 248?
- the text has been corrected
- The operational definition of the variable has not been presented. Also, since the survey item is not presented, it is impossible to determine the suitability of the results of this survey.
- the definitions of the variables and also the objectives of the survey were supplemented
- In the result of Part 5, the adoption and rejection of hypotheses are presented several times, which is very confusing. Please reorganize it in general to increase readability.
- the definitions of the variables and also the objectives of the survey were supplemented
- Conclusion part should separate the implication and limitation.
- Implications and limitations were added
Round 2
Reviewer 2 Report
The paper has updated according to reviewer's comments.
Reviewer 3 Report
It seems to be somewhat improved compared to before.